# Measuring the rate of NADPH consumption by glutathione reductase in the cytosol and mitochondria

Kenneth K. Y. Ting[1,2], Eric Floro[2,3], Riley Dow[1,2], Jenny Jongstra-Bilen[1,2,4], Myron I. Cybulsky[1,2,4,5]*, Jonathan V. Rocheleau[2,3,6,7]*

1 Department of Immunology, University of Toronto, Toronto, ON, Canada, 2 Toronto General Hospital Research Institute, University Health Network, Toronto, ON, Canada, 3 Institute of Biomedical Engineering, University of Toronto, Toronto, ON, Canada, 4 Department of Laboratory Medicine and Pathobiology, University of Toronto, Toronto, ON, Canada, 5 Peter Munk Cardiac Centre, University Health Network, Toronto, ON, Canada, 6 Department of Physiology, University of Toronto, Toronto, ON, Canada, 7 Banting and Best Diabetes Centre, University of Toronto, Toronto, ON, Canada

* myron.cybulsky@utoronto.ca (MIC); jon.rocheleau@utoronto.ca (JVR)

## Abstract

### Background

NADPH is an essential co-factor supporting the function of enzymes that participate in both inflammatory and anti-inflammatory pathways in myeloid cells, particularly macrophages. Although individual NADPH-dependent pathways are well characterized, how these opposing pathways are co-regulated to orchestrate an optimized inflammatory response is not well understood. To investigate this, techniques to track the consumption of NADPH need to be applied. Deuterium tracing of NADPH remains the gold standard in the field, yet this setup of mass-spectrometry is technically challenging and not readily available to most research groups. Furthermore, NADPH pools are compartmentalized in various organelles with no known membrane transporters, suggesting that NADPH-dependent pathways are regulated in an organelle-specific manner. Conventional methods such as commercial kits are limited to quantifying NADPH in whole cells and not at the resolution of specific organelles. These limitations reflect the need for a novel assay that can readily measure the consumption rate of NADPH in different organelles.

### Methods

We devised an assay that measures the consumption rate of NADPH by glutathione-disulfide reductase (GSR) in the mitochondria and the cytosol of RAW264.7 macrophage cell lines. RAW264.7 cells were transfected with Apollo-NADP+ sensors targeted to the mitochondria or the cytosol, followed by the treatment of 2-deoxyglucose and diamide. Intravital imaging over time then determined GSR-dependent NADPH consumption in an organelle-specific manner.

### Discussion

In lipopolysaccharide (LPS)-stimulated RAW264.7 cells, cytosolic and mitochondrial NADPH was consumed by GSR in a time-dependent manner. This finding was cross

**Data Availability Statement:** All relevant data are within the article and its Supporting Information files.

**Funding:** This work was supported by the Canadian Institutes of Health Research Grant FDN154299 (to M.I.C.). Additional support of this work was from Canadian Institutes of Health Research Project Grant DOL 409157 (to J.R.), and by Natural Sciences and Engineering Research Council of Canada Research Tools and Instrumentation Grant 2018-00846 (to J.R.). The funders had no role in study design, data collection and analysis, decision to publish, or preparation of the manuscript.

**Competing interests:** The authors have declared that no competing interests exist.

**Abbreviations:** Abbreviations, Definition; NADP$^+$, Nicotinamide adenine dinucleotide phosphate; GSR, Glutathione-disulfide reductase; GS-SG, Glutathione disulfide; GSH, Glutathione; ROS, Reactive oxygen species; LPS, Lipopolysaccharide; G6PD, Glucose-6-phosphate dehydrogenase; 2-DG, 2-deoxyglucose; oxLDL, Oxidized low-density lipoprotein; NRF2, Nuclear factor erythroid 2-related factor 2.

validated with a commercially available NADPH kit that detects NADPH in whole cells. Loading of RAW264.7 cells with oxidized low-density lipoprotein followed by LPS stimulation elevated GSR expression, and this correlated with a more rapid drop in cytosolic and mitochondrial NADPH in our assay. The current limitation of our assay is applicability to transfectable cell lines, and higher expression of plasmid-encoded sensors relative to endogenous glucose-6-phosphate dehydrogenase.

## Background

NADPH, the reduced form of nicotinamide adenine dinucleotide phosphate (NADP$^+$), is an essential co-factor for enzymes that catalyze a wide variety of oxidative and antioxidative processes [1]. Notably in myeloid cells, the importance of NADPH is indisputable for driving their effector functions. For instance, upon stimulation of myeloid cells, NADPH is significantly depleted due to consumption by inflammatory and anti-inflammatory pathways [2–5]. Although these individual pathways have been traditionally well-defined, how NADPH-dependent inflammatory and anti-inflammatory processes co-regulate each other to fine-tune an optimized inflammatory response is not well understood. Recently, we proposed that the competition for NADPH consumption between inflammatory and anti-inflammatory pathways is a naturally evolved mechanism that regulates the magnitude of an inflammatory response in immune cells [6]. To test this hypothesis, the consumption of NADPH needs to be accurately tracked and determined. Although the use of targeted metabolite tracing, such as deuterium tracing of NADPH [7], is successful and remains the gold standard in the field, this approach requires mass-spectrometry, which may not be accessible to many laboratories. More importantly, conventional commercial kits that quantify NADPH and NADP$^+$ only measure the steady-state total abundance and do not provide information on the direction of the metabolic flux (i.e., the rates of consumption vs. synthesis). Therefore, taken together, these limitations reflect the need for developing a more accessible and simpler assay that can measure the rate of NADPH consumption by a specific pathway, such as the one mediated by glutathione-disulfide reductase (GSR).

GSR is an anti-oxidative/anti-inflammatory enzyme that reduces glutathione disulfide (GS-SG) to glutathione (GSH) in a NADPH-dependent manner. GSR is conserved across eukaryotes and prokaryotes, with the exception of Gram-positive bacteria, some Gram-negative bacteria, certain insects, and parasites [8–11]. This conservation is not surprising as oxidative stress is an inevitable byproduct of aerobic respiration, and GSH is an important antioxidant that mitigates the effects of reactive oxygen species (ROS), such as hydrogen peroxide. As expected, GSR tends to be localized in regions of the cell where a high electron flux is located [12]. For instance, in prokaryotes, GSR is found in the periplasm [13,14], while in eukaryotes, it is found in the mitochondria, cytosol, endoplasmic reticulum and nucleus [15–17]. Thus, GSR is critical for regenerating GSH and maintaining a favorable redox environment. GSR enzymatic activity requires NADPH as a co-factor, which consumes NADPH by converting it to NADP$^+$. Since the consumption of NADPH by GSR may compete with NADPH-dependent inflammatory processes, we endeavored to determine NADPH consumption by GSR in macrophages (Mφs) stimulated by lipopolysaccharide (LPS), a classical Toll-like receptor 4 ligand.

Here, we describe a novel kinetic assay to measure NADPH consumption rate by GSR in the cytosol and the mitochondria (Fig 1A). This technique has been validated and applied in

our recently published study [6]. Because accurate determination of the NADPH consumption rate can be confounded by the reduction rate of NADP$^+$, we inhibited key biochemical pathways that reduce NADP$^+$ back to NADPH in LPS-stimulated Mφs, including glucose-6-phosphate dehydrogenase (G6PD), isocitrate dehydrogenases and folate-mediated one carbon metabolism that can contribute to the reduction of NADP$^+$ [18–20]. We added 2-deoxyglucose (2-DG) in our kinetic assay because it inhibits the early steps of glycolysis (hexokinase) [21] and subsequently downstream reactions, including the pyrophosphate pathway and the tricarboxylic acid cycle. We employed diamide, an established thiol-oxidizing agent that oxidizes GSH to GS-SG [22], to induce GSR-mediated NADPH consumption during reduction of GS-SG back to GSH (Fig 1A). By adding diamide and 2-DG simultaneously to LPS-stimulated cultured Mφs and measuring NADPH over a defined time interval, we can assess NADPH consumption by GSR in a kinetic assay.

As was described above, GSR is localized in different subcellular compartments and may differentially regulate NADPH levels. Conventional commercial kits that measure NADPH cannot provide organelle-specific information, which is an important limitation. To address

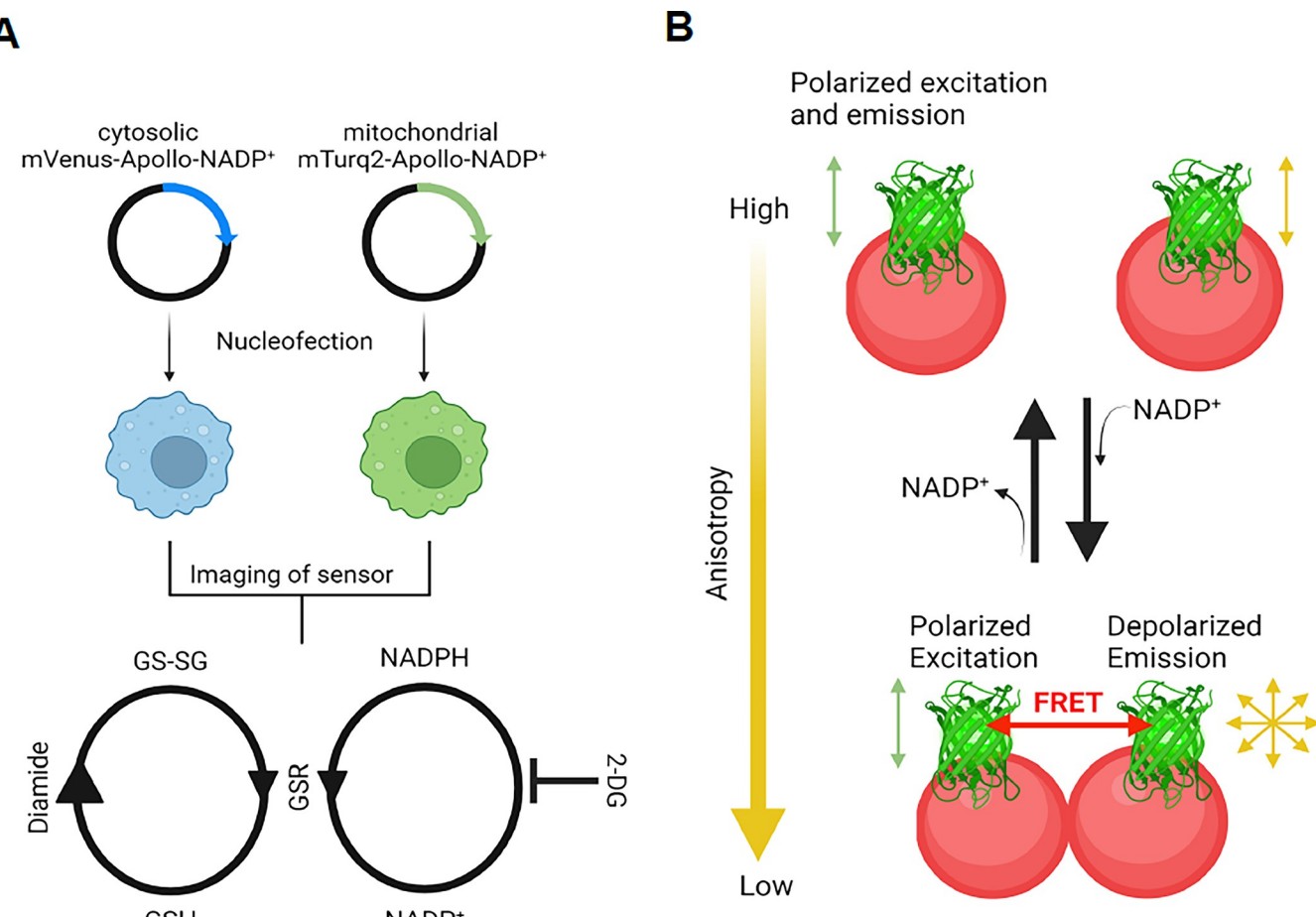

**Fig 1. Schematic of the assay.** (**A**) The entire workflow is illustrated. The assay begins with nucleofection of cytosolic mVenus-Apollo-NADP$^+$ and mitochondrial mTurq2-Apollo-NADP$^+$ sensors into RAW264.7 Mφ cell line, followed by imaging of the sensors. In a kinetic assay, 2-deoxyglucose (2-DG) and diamide are added together prior to an imaging time series. The addition of 2-DG blocks biochemical pathways that reduce NADP$^+$ to NADPH in LPS-stimulated cells. The addition of diamide oxidizes GSH to GS-SG, thereby stimulating the activity of GSR to reduce GS-SG back to GSH in a NADPH-dependent manner. (**B**) The mechanism of the Apollo-NADP$^+$ sensor is illustrated. The sensor, which is based on an enzymatically inactivated human G6PD tagged with a fluorescent protein, undergoes homodimerization in response to changes in free NADP$^+$ levels. This NADP$^+$-dependent dimerization/monomerization modulates FRET between the homologous fluorescent proteins (homoFRET) resulting in changes in steady-state fluorescence anisotropy that can then be used to assess the redox state of NADP$^+$/NADPH. (Created with Biorender.com).

this, we integrated organelle specific NADP$^+$ sensors into our assay. The Apollo-NADP$^+$ sensors are fluorescently tagged enzymatically inactive G6PD constructs that enable NADPH/NADP$^+$ redox states measurement through fluorescence anisotropy (Fig 1B) [23]. Recently, Chang et al improved the sensors by adding sequences that target protein expression to specific organelles, such as mitochondria [24]. Therefore, by transfecting Mφ cell lines with targeted Apollo-NADP$^+$ sensors and treating cells with 2-DG and diamide, we can measure NADPH consumption by GSR in the cytosol or mitochondria in a kinetic assay.

## Materials and methods

The protocol described in this peer-reviewed article is published on protocols.io, https://dx.doi.org/10.17504/protocols.io.rm7vzjoy4lx1/v1 and is included for printing as S1 File with this article.

### Biological materials

1. RAW264.7 Mφ cell line (ATCC, Catalog #:TIB-71)

### Reagents

1. 0.25% Trypsin/ 0.1% EDTA (Wisent, Catalog#:325–043 EL)

2. DMEM 1X with 4.5 g/L Glucose, L-Glutamine & Sodium Pyruvate (Wisent, Catalog#:319-005-CL)

3. Premium Fetal Bovie Serum (FBS) Endotoxin < 1 EU/ml (Wisent, Catalog#:091–150)

4. PBS 1X, w/o Calcium & Magnesium (Wisent, Catalog#:311–010 CL)

5. Geneticin™ Selective Antibiotic (G418 Sulfate) (50 mg/mL) (ThermoFisher, Catalog#: 10131035)

6. Amaxa® Cell Line Nucleofector® Kit V (LONZA, Catalog#: VCA-1003)

7. HBSS, no calcium, no magnesium, no phenol red (HBSS) 1X (ThermoFisher, Catalog#: 14175095)

8. Diamide (Sigma Aldrich, Catalog#: D3648)

9. 2-DG (Sigma Aldrich, Catalog#: D8375)

10. Mitochondrial mTurq2-Apollo-NADP+ (Addgene, Catalog#: 193303)

11. Cytosolic mVenus-Apollo-NADP+ (Addgene, Catalog#: 71797)

12. Fluorescein (Sigma Aldrich, Catalog#: 46955)

13. BMHH buffer (125 mm NaCl, 5.7 mm KCl, 2.5 mm CaCl2, 1.2 mm MgCl2, 10 mm HEPES, and 0.1% bovine serum albumin, pH 7.4)

14. Trypan Blue Solution (Sigma, Catalog#: T8154)

### Solutions and recipes

1. BMHH imaging buffer (125 mM NaCl, 5.7 mM KCl, 2.5 mM CaCl2, 1.2 mM MgCl2, 0.1% bovine serum albumin (BSA), and 10 mM 4-(2-hydroxyethyl)-1- piperazineethanesulfonic acid (HEPES) at pH 7.4).

## Laboratory supplies and equipment

1. Hemocytometer cell counting chamber (Sigma, Catalog#: BR719505-1EA)

2. Lonza™ Nucleofector™ Transfection 2b Device (Fisher Scientific, Catalog#: 13458999)

3. T75 tissue culture treated flask (StemCell Technologies, Catalog#: 200–0501)

4. T25 tissue culture treated flask (Sigma, Catalog#: C6356-200EA)

5. 35 mm Dish (No. 1.5 Coverslip, 14 mm Glass Diameter, Uncoated) (Mattek, Catalog#: P35G-1.5-14-C)

6. Custom-built wide-field RAMM microscope (ASI) equipped with excitation light-emitting diodes (LEDs) (405, 505, and 590 nm) and an excitation polarizing filter (Edmund Optics, Barrington, NJ), and built to perform simultaneous two-color fluorescence anisotropy imaging.

7. Images were collected using 40×/0.75 NA air objective lens (Olympus, Richmond Hill, Canada).

8. Fluorescence was passed through a Cerulean (ET470/24M) or Venus (ET535/30M) emission filter on a filter wheel, and subsequently split using an Optosplit II (Cairn, Faversham, U.K.) to simultaneously collect parallel and perpendicular emission light on separate regions of an IRIS 15 CMOS camera (Teledyne Photometrics, Tucson, AZ). Excitation was also passed through a triple band filter containing ET-350/55-470/30-557/35 band-pass filters.

## Procedures

### Part I: Transfection of Apollo-NADP$^+$ sensors in RAW264.7 Mφ cell lines

One week prior to nucleofection, passage cells in 15mL of DMEM media supplemented with 10% FBS (growth media) every 2–3 days in a T75 tissue culture treated flask.

1. To passage cells, remove cultured media and treat cells with 3 mL of trypsin for 5 minutes at 37˚C/5% $CO_2$ for detachment. Next, add 3 mL of fresh DMEM media with 10% FBS to neutralize trypsin and centrifuge the cells at 90xg for 5 minutes (22˚C).

2. Remove the supernatant, resuspend the cells with 10mL of fresh DMEM media, quantify the number of cells using a hemocytometer cell counting chamber and passage them at a ratio of 1:5 to 1:8.

3. One day prior to nucleofection, passage the cells at a ratio of 1:3.

4. On the day of nucleofection, retrieve the nucleofector solution and supplement from Amaxa® Cell Line Nucleofector® Kit V. For the first time usage, ensure to add all the supplements to the solution.

5. Fill one well of a 6-well plate with 1 mL of DMEM that is supplemented with 20% FBS. Pre-warm the plate at 37˚C/5% $CO_2$.

6. Centrifuge and isolate $2 \times 10^6$ cells. Remove supernatant and rinse cells with 1mL of pre-warmed 1X PBS. Repeat centrifugation, remove PBS and resuspend pellet with 100 μl of supplemented nucleofector solution. Add 5 μg of mitochondrial mTurq2-Apollo-NADP$^+$ sensor or cytosolic mVenus-Apollo-NADP$^+$ sensor to the mixture.

7. Transfer the complete mixture to kit-supplied cuvette without the injection of bubbles. Insert the cuvette into a Nucleofector® II Device and select program D-032 by pressing the X-button. Upon completion of nucleofection, retrieve the cuvette and add 500 µl of pre-warmed DMEM supplemented with 20% FBS into the suspension.

8. Transfer the entire 600ul of suspension using kit-supplied pipettes into the pre-warmed 6 well-plate in a drop-wise manner (total volume is 1.6 mL in the well).

9. 3 hours after the cells have adhered, replace the cultured media with 2mL of fresh DMEM media supplemented with 20% FBS for recovery.

10. When cell confluency has reached a minimum of 80% (usually 48 hours post transfection), the cells can be replated for experiments where transient nucleofection of the sensor is desired.

11. For stable selection, re-seed all the transfected cells from step 11 to a T25 culture flask with 5mL of DMEM media supplemented with 10% FBS. 3 hours after the cells have adhered, replace the cultured media with fresh DMEM media supplemented with 10% FBS and G418 Sulfate at final concentration of 400 µg/ml.

12. Replace the media with fresh 5mL of DMEM media supplemented with 10% FBS and G418 for every 2–3 days until one week. If cell confluency has reached a minimum of 80% in the T25 flask, re-seed the cells to a T75 flask or 10 cm tissue culture treated dish with 10mL of DMEM media supplemented with 10% FBS and G418.

13. To set up imaging of the sensors, isolate and seed $1 \times 10^6$ cells in a series of 35-mm petri dish (one dish per time point of the kinetic) with 2mL of DMEM media supplemented with 10% FBS and G418 one day prior to imaging.

## Part II: Imaging of the Apollo-NADP$^+$ sensors in RAW264.7 Mφ cell lines

On the day of imaging, prepare a 1:1 mixture of diamide (250 mM) and 2-DG (250 mM) in HBSS buffer. Specifically, add 1mL of 500 mM of diamide with 1mL of 500 mM of 2-DG.

1. On the day of imaging, prepare and calibrate microscope for imaging of Apollo-NADP + sensor.

    a. Turn on microscope at least 30min prior to any imaging to ensure CMOS camera temperature reaches equilibrium (Iris 15 camera was used in this case).

    b. Launch microscope management software, in this case open source software Micro-Manage was used (available for download at https://micro-manager.org/Download_Micro-Manager_Latest_Release)

        i. Set binning. In this case binning was set to 4x4 to collect the maximum image intensity.

        ii. Live imaging parameters should be set low to limit photobleaching and phototoxicity. In this case LED and exposure time were set to 20% and 100 ms, respectively.

    c. Ensure LED is properly aligned (i.e., evenly illuminates the back aperture of the lens) by first turning on the YFP LED, then holding a piece of paper against the microscope stage adjust the LED position until the emitted light appearing on the paper becomes sharp.

    d. Calibrate polarizer by first ensuring the polarizer is placed in the excitation path.

        i. Next, place a YFP-autofluorescent-plastic-slide on the microscope stage.

ii.   Begin live imaging at low intensity (20%) and low exposure time (10–50 ms), then using the draw tool draw a box around the live image of both the parallel and perpendicular channels.

iii.   Then select 'Analyze' > 'Plot Profile'. This will display a live profile of the intensity of both the parallel and perpendicular channels.

iv.   Finally, adjust the polarizer until the difference in intensity values between the parallel and perpendiculr channels is maximized. When done correctly the parallel channel will have an overall higher intensity than perpendicular.

2. Remove the cultured media and replace it with 900µl of pre-warmed HBSS.

3. Determine image acquisition parameters, LED intensity (%) and exposure time (ms).

   a. Place cells in pre-warmed HBSS onto stage top and use brightfield settings locate and focus on the cells.

   b. Switch to fluorescence imagine (e.g., YFP imaging), and use live imaging settings to fine focus on the cells/sensor. Set LED intensity to 80% and experiment with different image acquisition exposure times to find parameters that produce final sensor intensities within your camera's known saturation levels. In this case the total intensity in the parallel channel was roughly 40,000 AU, which was commonly achieved with an LED intensity of 80% and exposure time of 800 ms.

4. For the first time point, do not add any diamide and 2-DG. Image the sensor to acquire the baseline value of the control and experimental condition.

   a. TROUBLESHOOTING: Please refer to *General Notes and Troubleshooting* if the sensors were not expressed or weakly expressed.

5. For the second time point, add 100µl of the 1:1 mixture of diamide and 2-DG into the first petri dish, such that the final concentration of diamide and 2-DG is 25mM each. Incubate for 5 minutes, followed by imaging of the sensors.

6. For the third and subsequent time points, repeat step 6 for an additional 5 minute-interval until the completion of the kinetics.

7. At the end of imaging acquire images of standard anisotropy fluorescein solution for instrument diagnostics and G-factor correction.

   a. Start by preparing a solution of 5 mM fluorescein dissolved in BMHH.

   b. Set LED intensity and exposure time such that the acquired image has roughly the same total intensity of the sensor in cells, in this case 40,000 AU at 20% LED and 20 ms exposure time.

   c. Fluorescein solution can be left in the fridge at 4˚C for 1 month for repeated uses.

## Part III: Data analysis

Parallel ($I\|$) and perpendicular ($I\perp$) fluorescence intensity images were analyzed with a custom ImageJ plugin. These are available at https://github.com/RocheleauLab/Optosplit-Anisotropy-Analysis-scripts. The following three imageJ macro scripts are required:

   a. 00-Preporcessing

b. 01-Anisotropy Macro

c. 03-ROI Selection

2. The images were background-corrected using a rolling ball filter where the radius is larger than the largest cell in your images but small enough that distortions in the background are removed, in this case a radius of 100 was used. Pixel-by-pixel anisotropy (r) was calculated using the background-corrected intensities: $r = (I\| - GI\perp)/(I\| + 2G\perp)$.

a. Start by navigating to the folder containing all images and create the following two folders:

 i. GfactorImages

 ii. AlignmentImages

b. Drag and drop your fluorescein image into the GfactorImages folder.

c. Select an image of your cells and drag and drop into the imageJ application.

 i. On the imageJ application select Plugins > Macros > Record...

 ii. Using the draw Rectangle tool, draw two boxes of the same size and position on the parallel and perpendicular channels of the image

 iii. Dimensions for the boxes drawn will appear in the Record dialogue box. Copy and paste these dimensions into 00-Preprocessing script under rect_para and rect_perp

 iv. Run 00-Preprocessing script

 v. A file explore dialogue box will open. Navigate to Alignment Folder > Transform "select"

 vi. Another file explore dialogue box will open. This time navigate to Alignment Folder > Original > and "select" the image titled "Para"

 vii. The script will continue running through all the images

 viii. Processed images will be placed in a new folder titled "T2_Processed"

 ix. The script will continue running through all images in the experiment folder. This may take several minutes and depends on the number of images acquired

3. Gfactor for the wide-field microscope was calculated using fluorescein solutions, which have an anisotropy near zero, thus simplifying the standard anisotropy equation to $G = I\|/I\perp$.

a. After 00-Preprocessing script has finished running, navigate to 01-Anisotropy Macro script and press Run

b. A file explorer dialogue box will open, select the experiment folder you are working with

c. Allow the script to run. This may take several minutes depending on the number of images acquired

4. Regions of interest (ROI) were selected using the parallel intensity images to avoid selection bias.

a. After 01-Anisotropy Macro script has finished, navigate to 03-ROI Selection and press Run

b. Using either the oval tool or the 'any shape tool' draw regions of interest over the largest portion of where the sensor is appearing in the Para channel

5.  Each independent trial consists of at least five to more than 30 cells (5–30 technical replicates).

Other procedures are described below.

*Immunoblotting*

RAW264.7 cells were cultured in 12-well plates at 2 x $10^6$ cells per well/experimental condition. Cells were lysed for 15 minutes in ice-cold RIPA buffer consisting of 1% NP40, 0.1% SDS, 0.5% deoxycholate in PBS, supplemented with 1 mM PMSF, 1X cOmplete™, EDTA-free Mini Protease Inhibitor Cocktail (Sigma Cat#11873580001) and 1X PhosSTOP™ (Sigma Cat#4906845001). Protein concentrations in lysates were determined by Protein Assay Dye Reagent (BioRad #5000006), diluted in 2x Laemmli sample buffer (BioRad Cat#161–0737) with fresh β-mercaptoethanol (BioRad #1610710), and heated at 95˚C for 5 minutes. Samples (20 μg of protein per lane) were resolved on 8% SDS-PAGE gels and transferred to polyvinylidene difluoride membranes (Sigma #IPVH00010) using a wet transfer system. Membranes were blocked with 5% skim milk non-fat powder in Tris-buffered saline-Tween (TBST) for 1 h at room temperature. Membranes were incubated with primary antibodies diluted in 5% milk 1X TBST overnight at 4˚C: anti-Actin (1:3000) (Sigma, A2066), anti-NRF2 (1:1000) (Cell Signaling Technology #12721) and anti-GSR (1:1000) (ThermoFisher, PA5-29945), followed by washing and incubation with HRP-conjugated anti-rabbit IgG (1:2000) (CST#7074) in 5% milk 1X TBST (22˚C, 1 h). Blots were developed using Immobilon Forte Western HRP substrate (Sigma, WBLUF0100) and imaged with Microchemi 4.2 (BioRad). Original uncropped blots are provided in S2 File with this article.

Quantification of total NADPH levels

Total NADPH assay kits (Abcam, #ab186031) were used. In brief, cells were cultured in 6-well plates, then loaded with oxLDL for 24 h with subsequent 6 h LPS stimulation the next day. Cells were then lysed with respective lysis buffers provided in the kits and the abundance of metabolites was determined according to the manufacturers' protocols.

RNA isolation and real-time quantitative PCR

Total RNA was isolated with E.Z.N.A.® Total RNA Kit I (Omega Cat#R6834-01) and reverse transcription (RT) reactions were performed with High-Capacity cDNA Reverse Transcription Kit (ThermoFisher Cat#4368814) according to manufacturer's protocol. Real-time quantitative-PCR (qPCR) was then performed using Roche LightCycler 480 with Luna® Universal qPCR Master Mix (New England Biolabs, Cat#M3003E). Quantification of mRNA was performed by using primers that span over two adjacent exons, quantified using the comparative standard curve method and normalized to hypoxanthine phosphoribosyltransferase (*Hprt*), a housekeeping gene. Primer sequences used for qPCR are listed below:

*Gsr* Fwd: GTTTACCGCTCCACACATCCTG Rev: GCTGAAAGAAGCCATCACTGGTG

*Gclm* Fwd: TCCTGCTGTGTGATGCCACCAG Rev: GCTTCCTGGAAACTTGCCTCAG

*Nqo1* Fwd: GCCGAACACAAGAAGCTGGAAG Rev: GCAAATCCTGCTACGAGCACT

*Srxn1* Fwd: TACCAATCGCCGTGCTCATCCG Rev: CCTTTGATCCAGAGGACGTCGA

*Hmox1* Fwd: CACTCTGGAGATGACACCTGAG Rev: GTGTTCCTCTGTCAGCATCACC

*Hprt* Fwd: CAAGCTTGCTGGTGAAAAGGA Rev: TGAAGTACTCATTATAGTCAAGGGCATATC

The volume of each reagent used in a 20 ul qPCR reaction is listed below:

The setting for the cycling time for the qPCR reaction is listed below:

| Component | 20 μl reaction | Final concentration |
| --- | --- | --- |
| Luna Universal qPCR master mix | 10 μl | 1X |
| Forward primer (10 μM) | 0.5 μl | 0.25 μM |
| Reverse primer (10 μM) | 0.5 μl | 0.25 μM |
| Template DNA | Variable | <100 ng |
| Nuclease-free water | Up to 20 μl | |

**Statistical analysis.** The statistical test(s) used in each experiment is listed in the figure

| Cycle step | Temperature | Time | Cycles |
|---|---|---|---|
| Initial denaturation | 95˚C | 60 seconds | 1 |
| Denaturation | 95˚C | 15 seconds | 40–45 |
| Extension | 60˚C | 30 seconds (+ plate read) | |
| Melt Curve | 60–95˚C | Various | 1 |

legends. In general, the figures show pooled data from independent experiments. All experiments were repeated at least three times, and the number of biological replicates is indicated as the n value. Statistical analyses were performed using the Prism software (9.2.0), unless otherwise specified in the figure legends.

## General notes and troubleshooting

Although the expression levels of cytosolic and mitochondrial Apollo-NADP$^+$ sensors were adequate in transiently (>48 h) and stably (>1 week) transfected RAW264.7 cells, we found the quantification of anisotropy to be less variable in stably transfected cells. Therefore, we recommend the use of stably transfected cells for experimental data acquisition, while transiently transfected cells can be used to verify targeting of specific organelles by different sensors.

Since the Apollo-NADP$^+$ sensor response depends on the interaction (dimerization) of exogenous mutant G6PD tagged with a fluorescent protein, interactions between transfected and endogenous G6PD may reduce sensor response. Therefore, in cell types with large amounts of endogenous G6PD, the apparent dynamic range of the sensor's response may be reduced. To confirm that the sensor responds in a particular cell type, diamide can be used to induce sensor dimerization and the magnitude of the response can be measured.

| Step | Problem | Possible reason | Solution |
|---|---|---|---|
| Imaging of cytosolic and/or mitochondrial Apollo-NADP$^+$ sensor post transient nucleofection. | No or weak expression of cytosolic and mitochondrial sensor post transient nucleofection. | Insufficient amount of plasmid was added during nucleofection. | 1. Increase the amount of plasmid (up to 5µg) per nucleofection.<br>2. Ensure the plasmid was properly extracted during plasmid preparation with no LPS contamination.<br>3. Sequence the plasmid to ensure the sequence matches with the original listed sequence posted on Addgene. |
| | | Incorrect time-window to detect the expression of plasmid. | Determine the best time-window to detect the expression of plasmid from 24h to 72h post transient nucleofection. |
| Imaging of cytosolic and/or mitochondrial Apollo-NADP$^+$ sensor post stable nucleofection. | No or weak expression of cytosolic and mitochondrial sensor post transient nucleofection. | Incorrect dosage of G418 was used to stably select transfected cells. | Perform a serial-dilution of G418 and determine the minimum concentration needed to kill non-transfected cells within one week. |
| | | The weakly expressed clones were outcompeting the strongly expressed clones. | Perform clonal selection and expansion post stable nucleofection. Characterize all the isolated clones and select the strongly expressed ones for future experiments. |
| Drawing regions of interest (ROI) during analysis. | Anisotropy values vary with certain regions of the cell. | Cell nucleus excludes the sensor and therefore does not exhibit valid anisotropy values. | When drawing the ROI's ensure that the region is within the cell boundary and does not include the nucleus. |
| | | Edges of the cell express unreliable anisotropy values | |

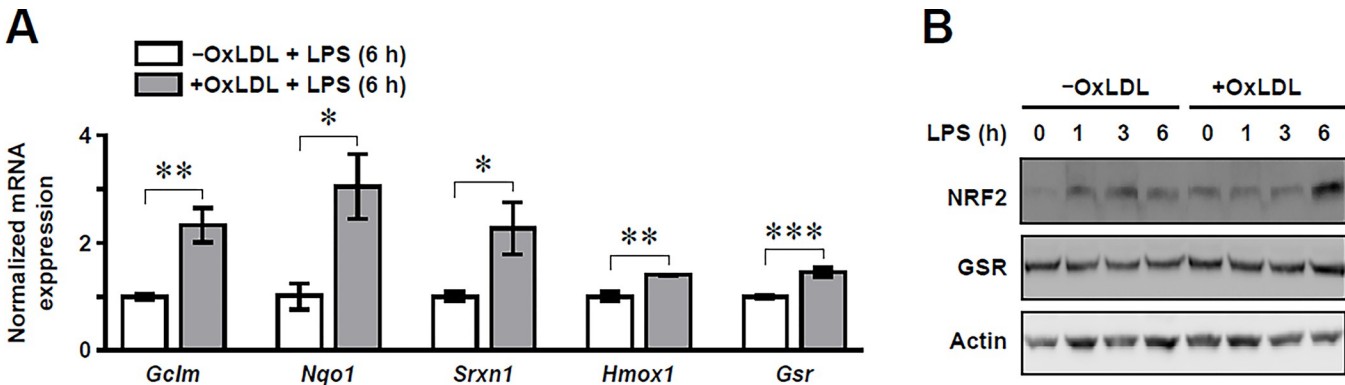

**Fig 2. OxLDL loading of RAW264.7 Mφs upregulates NRF2-dependent transcriptional regulation post LPS stimulation. (A)** qPCR data showing the normalized mRNA expression of NRF2-regulated genes (*Gclm*, *Nqo1*, *Srxn1*, *Hmox1*, *Gsr*) in RAW264.7 Mφs with (+) and without (-) oxLDL loading and LPS stimulation for 6 h (n = 4–5). Values are normalized to the group without oxLDL loading. The mean ± SEM is plotted. Significant differences were determined by an unpaired Student's *t*-test (*P<0.05, **P<0.01, ***P<0.001). **(B)** Representative immunoblots showing NRF2, GSR and actin protein expression in RAW264.7Mφs with and without oxLDL loading. A LPS stimulation time course (0 to 6 h) was performed.

## Expected results

Prior to the validation of our assay, we increased the expression of GSR by loading RAW264.7 Mφs with oxidized low-density lipoprotein (oxLDL) and then stimulating with LPS. This was based on our previous reports that incubating primary murine Mφs with oxLDL or cholesterol is a reproducible method of lipid loading Mφs, and this significantly upregulated NRF2-dependent transcriptional regulation after LPS stimulation [6,25,26]. These experimental conditions are important as increasing GSR protein expression will directly increase its activity and consumption of NADPH, thereby serving as a positive control for our assay. GSR is a NRF2-regulated gene [27]. Therefore, as expected we found increased mRNA expression of *Gsr* and other NRF2-regulated genes (*Gclm*, *Nqo1*, *Srxn1* and *Hmox1*) (Fig 2A), as well as NRF2 and GSR protein abundance (Fig 2B) in LPS-stimulated RAW264.7 Mφs with accumulated oxLDL.

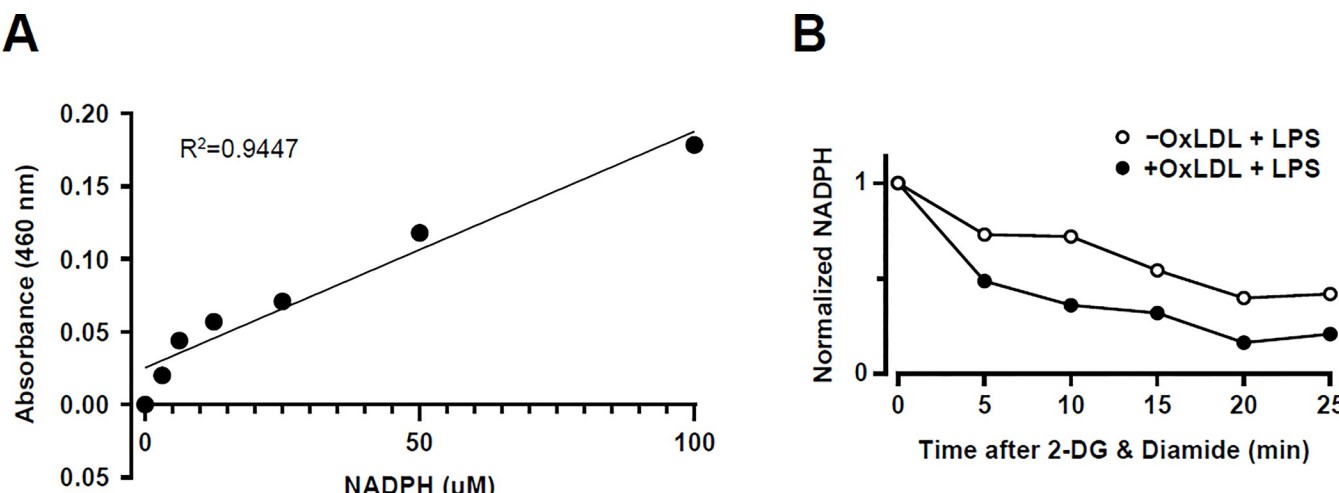

**Fig 3. OxLDL loading of RAW264.7 Mφs enhances GSR-dependent NADPH consumption 6 hours after LPS stimulation. (A)** Representative NADPH standard curve generated for analysis of RAW264.7 Mφs-derived lysates. **(B)** Representative experiment showing the quantification of NADPH abundance in RAW264.7Mφs with (+) and without (-) oxLDL loading and 6 h post-LPS stimulation. Samples were obtained at sequential time points after the addition of 2-deoxyglucose (2-DG) and diamide.

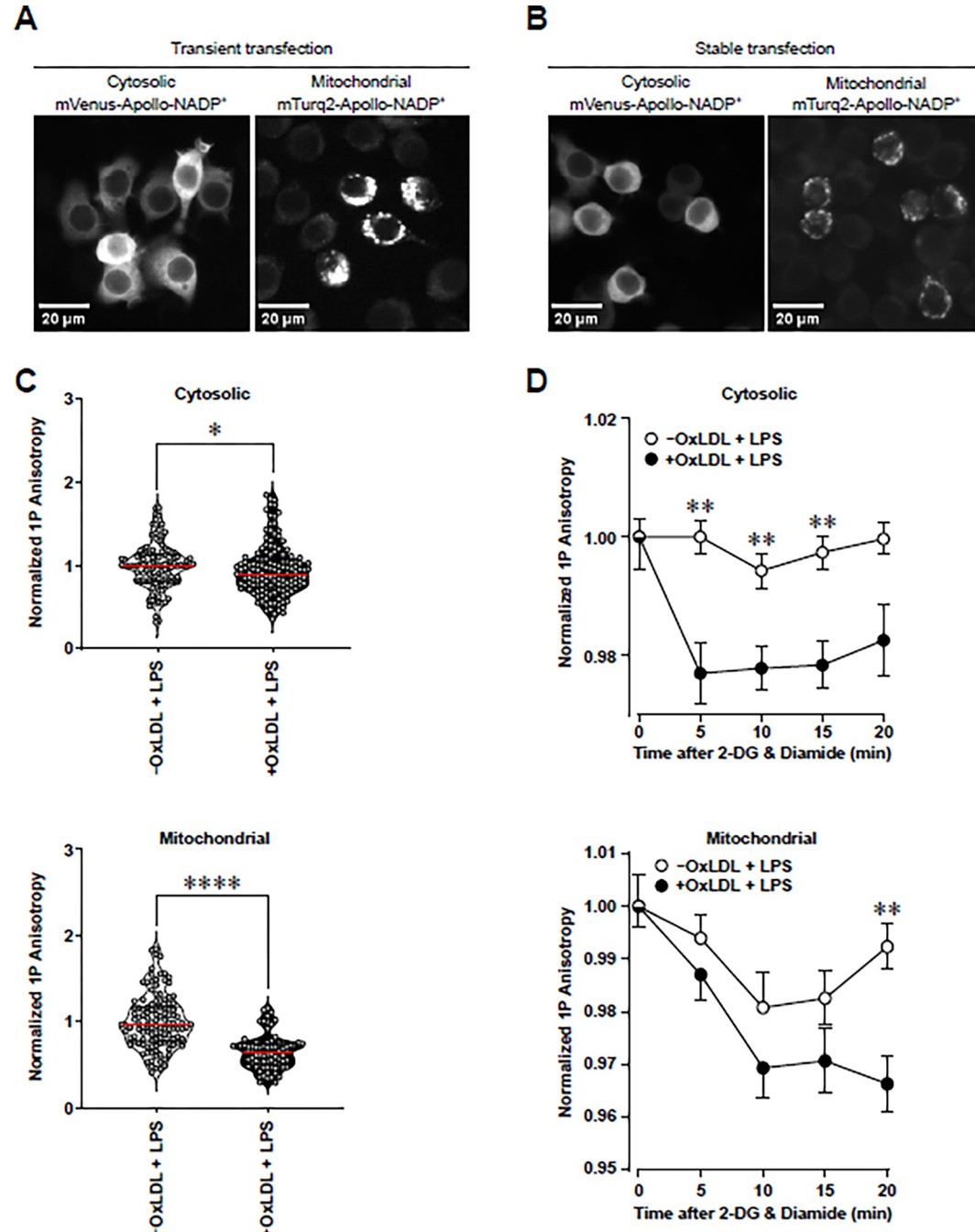

**Fig 4. OxLDL loading of RAW264.7 Mφs enhances GSR-dependent NADPH consumption in the mitochondria and the cytosol 6 hours after LPS stimulation. (A, B)** Representative microscopic images of cytosolic mVenus-Apollo-NADP+ and mitochondrial mTurq2-Apollo-NADP+ sensors 48 h after transient nucleofection (A) and 1 week after stable transfection (B). Magnification bars represent 20 μm. **(C)** 1P anisotropy quantification of cytosolic mVenus-Apollo-NADP+ and mitochondrial mTurq2-Apollo-NADP+ sensors in transiently-transfected RAW264.7 cells with (+) and without (-) oxLDL loading and 6 h of LPS stimulation (n = 3). **(D)** Representative kinetic assay showing quantification of 1P anisotropy of cytosolic mVenus-Apollo-NADP+ and mitochondrial mTurq2-Apollo-NADP+ sensors in stably-transfected RAW264.7 cells ±oxLDL loading, LPS stimulation (6 h) and treatment with 2-DG and diamide. The mean ± SEM is plotted in all graphs. Significant differences (*$P < 0.05$, **$P < 0.01$, ****$P < 0.0001$) were determined by an unpaired Student's $t$-test in (C) or by a two-way ANOVA with Bonferroni correction in (D).

These data confirm that oxLDL loading of RAW264.7 Mφs increases GSR abundance after LPS stimulation and thus justify the use of these conditions to validate our proposed assay.

We measured the rate of NADPH consumption by GSR in whole cells using a commercial kit that quantifies NADPH. The concentration of NADPH in a cell lysate is determined by the colorimetric absorbance after adding the supplied NADPH probe solution. After confirming that the concentration of NADPH in our lysates was within the linear detection range of the assay (Fig 3A), we treated LPS-stimulated (6 h) RAW264.7 Mφs with 2-DG and diamide and quantified NADPH kinetics by obtaining lysates at 5-minute intervals for a total of 25 minutes (Fig 3B). Whole cell NADPH was reduced in a time-dependent manner, suggesting that NADPH was being consumed by GSR. Notably, in oxLDL loaded Mφs, NADPH was reduced more rapidly. This is expected as GSR mRNA and protein expression is higher in these Mφs (Fig 2).

After confirming the feasibility of our proposed kinetic assay, we proceeded to integrate determination of NADPH in subcellular compartments using targeted Apollo-NADP$^+$ sensors. To do this, we first confirmed that both cytosolic- and mitochondrial-targeted Apollo-NADP$^+$ sensors are expressed in appropriate compartments of RAW264.7 Mφs. Transient transfection of cytosolic and mitochondrial Apollo-NADP$^+$ sensors was detected by fluorescence microscopy 48 h post-nucleofection (Fig 4A). One week of geneticin selection post-nucleofection resulted in stable expression of both sensors (Fig 4B). To test the functionality of the sensors, we cultured transiently transfected RAW 264.7 Mφs with or without oxLDL for 24 h, stimulated the cells with LPS for 6 h and performed imaging of the sensors. OxLDL loading significantly inhibited anisotropy in both the cytosol and mitochondrial matrix (Fig 4C), which indicates a higher NADP$^+$ to NADPH ratio. Finally, to determine if the reduction of NADPH to NADP$^+$ was due to increased consumption by GSR, we performed a kinetic assay using stably-transfected RAW 264.7 Mφs with or without oxLDL loading, LPS stimulation and the addition of 2-DG and diamide. OxLDL loading significantly depleted cytosolic and mitochondrial anisotropy in a time-dependent manner (Fig 4D), suggesting that oxLDL loading increased GSR-mediated NADPH consumption in both compartments.

## Supporting information

**S1 File. Step-by-step protocol, also available on protocols.io.**
(PDF)

**S2 File. Original uncropped immunoblots.**
(PDF)

## Author Contributions

**Conceptualization:** Eric Floro, Myron I. Cybulsky, Jonathan V. Rocheleau.

**Data curation:** Kenneth K. Y. Ting, Eric Floro.

**Formal analysis:** Kenneth K. Y. Ting, Eric Floro, Riley Dow.

**Funding acquisition:** Kenneth K. Y. Ting, Myron I. Cybulsky, Jonathan V. Rocheleau.

**Investigation:** Kenneth K. Y. Ting, Eric Floro, Riley Dow.

**Methodology:** Kenneth K. Y. Ting.

**Software:** Jonathan V. Rocheleau.

**Supervision:** Jenny Jongstra-Bilen, Myron I. Cybulsky, Jonathan V. Rocheleau.

**Visualization:** Kenneth K. Y. Ting, Eric Floro.

**Writing – original draft:** Kenneth K. Y. Ting, Eric Floro, Jonathan V. Rocheleau.

**Writing – review & editing:** Kenneth K. Y. Ting, Eric Floro, Myron I. Cybulsky, Jonathan V. Rocheleau.

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
