## [Decision Letter · Decision Letter 0]

18 Jul 2024

PONE-D-24-18553Measuring the rate of NADPH consumption by glutathione reductase in the cytosol and mitochondriaPLOS ONE

Dear Dr. Cybulsky,

Thank you for submitting your manuscript to PLOS ONE. After careful consideration, we feel that it has merit but does not fully meet PLOS ONE’s publication criteria as it currently stands. Therefore, we invite you to submit a revised version of the manuscript that addresses the points raised during the review process. 

A marked-up copy of your manuscript that highlights changes made to the original version. You should upload this as a separate file labeled 'Revised Manuscript with Track Changes'.An unmarked version of your revised paper without tracked changes. You should upload this as a separate file labeled 'Manuscript'.If applicable, we recommend that you deposit your laboratory protocols in protocols.io to enhance the reproducibility of your results. Protocols.io assigns your protocol its own identifier (DOI) so that it can be cited independently in the future. For instructions see: https://journals.plos.org/plosone/s/submission-guidelines#loc-laboratory-protocols. Additionally, PLOS ONE offers an option for publishing peer-reviewed Lab Protocol articles, which describe protocols hosted on protocols.io. Read more information on sharing protocols at https://plos.org/protocols?utm_medium=editorial-email&utm_source=authorletters&utm_campaign=protocols.

We look forward to receiving your revised manuscript.

Kind regards,

Chen-Guang Liu, Ph.D.

Academic Editor

PLOS ONE

2. PLOS requires an ORCID iD for the corresponding author in Editorial Manager on papers submitted after December 6th, 2016. Please ensure that you have an ORCID iD and that it is validated in Editorial Manager. To do this, go to ‘Update my Information’ (in the upper left-hand corner of the main menu), and click on the Fetch/Validate link next to the ORCID field. This will take you to the ORCID site and allow you to create a new iD or authenticate a pre-existing iD in Editorial Manager. Please see the following video for instructions on linking an ORCID iD to your Editorial Manager account: https://www.youtube.com/watch?v=_xcclfuvtxQ.

Additional Editor Comments:

Dear Dr. Myron Cybulsky

Thank you for sending the manuscript "Measuring the rate of NADPH consumption by glutathione reductase in the cytosol and mitochondria" to PLOS ONE for publication.

Given the reviewer's comments, minor issues still need to be solved before acceptance. I have consolidated the review comments below, hoping they will help you modify the paper.

Reviewer 1:

I read with great interest the authors' manuscript entitled "Measuring the Rate of NADPH Consumption by Glutathione Reductase in the Cytosol and Mitochondria," which is intended for submission to PLOS ONE. Congratulations on this publication; it is a very thorough and valuable contribution. Since this is a laboratory protocol, my research team and I reviewed it to identify any questions we might have if we were to perform this measurement. We have only a few technical questions, which I have noted in the attached PDF.

Reviewers' comments:

Reviewer's Responses to Questions

**Comments to the Author**

1. Does the manuscript report a protocol which is of utility to the research community and adds value to the published literature?

Reviewer #1: Yes

2. Has the protocol been described in sufficient detail?

To answer this question, please click the link to protocols.io in the Materials and Methods section of the manuscript (if a link has been provided) or consult the step-by-step protocol in the Supporting Information files.

The step-by-step protocol should contain sufficient detail for another researcher to be able to reproduce all experiments and analyses.

Reviewer #1: Yes

3. Does the protocol describe a validated method?

Reviewer #1: Yes

4. If the manuscript contains new data, have the authors made this data fully available?

Reviewer #1: N/A

**5. Is the article presented in an intelligible fashion and written in standard English?**

Reviewer #1: Yes

6. Review Comments to the Author

Reviewer #1: I read with great interest the authors' manuscript entitled "Measuring the Rate of NADPH Consumption by Glutathione Reductase in the Cytosol and Mitochondria," which is intended for submission to PLOS ONE. Congratulations on this publication; it is a very thorough and valuable contribution. Since this is a laboratory protocol, my research team and I reviewed it to identify any questions we might have if we were to perform this measurement. We have only a few technical questions, which I have noted in the attached PDF.

Thank you for the opportunity to review the article.

7. PLOS authors have the option to publish the peer review history of their article (what does this mean?). If published, this will include your full peer review and any attached files.

Reviewer #1: **Yes: **Petra Hartmann

---

## [Author Response · Author response to Decision Letter 0]

31 Jul 2024

RESPONSE TO THE REVIEWERS’ COMMENTS

I would like to thank reviewer #1 for their insightful comments (copied below in bold font). Each point that was raised is addressed below. Changes to the text in the revised manuscript are highlighted in yellow. 

1. Based on what did you use 10 ml of trypsin (usually a lot for a flask of this size)? Please describe this. What is the final concentration of trypsin? Is it necessary to cool the fugue to this cell line?

We agree with the reviewer that 10 ml of trypsin is more than the typical recommended volume for a T75 flask. We have now adjusted the volume of both trypsin and DMEM, which deactivates the trypsin, to 3 ml (LINE 164 – 165). This volume is based on the recommendations from StemCell Technologies as we purchased the T75 flasks from this company (https://cdn.stemcell.com/media/files/pis/29301-PIS_1_0_2.pdf). We believe that changing the volume of trypsin will not change the results generated in our study. The final concentration of trypsin is 0.25% Trypsin/ 0.1% EDTA prior to deactivation with 10% FBS in DMEM. 

We are unsure of the last question. We assume that the reviewer is asking if it is necessary to cool the centrifuge for this cell line. Our response is that centrifugation was performed at room temperature or 22 degrees C (LINE 165).

2. What conditions were used during this time?

We have now revised the manuscript and provided the recommended antibody concentration dilution in our method section, as well as the temperature of primary antibody incubation with PVDF membrane overnight (LINE 330 – 333). 

3. Have you used a standard protocol for QPCR ? If not, please provide details of the final concentrations.

Yes. We have used a standard protocol for QPCR. We have now revised the manuscript to provide more details of the final concentration and the QPCR protocol (LINE 356 - 358).

4. Please clarify the item numbers of groups. Only one group has an item number description. Please specify the item numbers of the groups. What statistical software used to? Please indicate this in the main text.

We have now revised the manuscript and provided a paragraph in the method section, titled as “statistical analysis”, to clarify the definition of ‘n’ and the description of statistical software used to calculate the provided statistics (LINE 360 – 364).

---

## [Decision Letter · Decision Letter 1]

21 Aug 2024

Measuring the rate of NADPH consumption by glutathione reductase in the cytosol and mitochondria

PONE-D-24-18553R1

Dear Dr. Cybulsky,

We’re pleased to inform you that your manuscript has been judged scientifically suitable for publication and will be formally accepted for publication once it meets all outstanding technical requirements.

Kind regards,

Chen-Guang Liu, Ph.D.

Academic Editor

PLOS ONE

Additional Editor Comments (optional):

Dear Dr. Myron Cybulsky

I am glad to inform you that the manuscript "Measuring the rate of NADPH consumption by glutathione reductase in the cytosol and mitochondria" could be accepted by PLOS ONE for publication.

Reviewers' comments:

Reviewer's Responses to Questions

**Comments to the Author**

1. Does the manuscript report a protocol which is of utility to the research community and adds value to the published literature?

Reviewer #1: Yes

2. Has the protocol been described in sufficient detail?

To answer this question, please click the link to protocols.io in the Materials and Methods section of the manuscript (if a link has been provided) or consult the step-by-step protocol in the Supporting Information files.

The step-by-step protocol should contain sufficient detail for another researcher to be able to reproduce all experiments and analyses.

Reviewer #1: Yes

3. Does the protocol describe a validated method?

Reviewer #1: Yes

4. If the manuscript contains new data, have the authors made this data fully available?

Reviewer #1: Yes

**5. Is the article presented in an intelligible fashion and written in standard English?**

Reviewer #1: Yes

6. Review Comments to the Author

Reviewer #1: The authors have answered the questions raised and I have no objections to the publication of this manuscript.

7. PLOS authors have the option to publish the peer review history of their article (what does this mean?). If published, this will include your full peer review and any attached files.

Reviewer #1: **Yes: **Petra Hartmann

---

## [Editor Report · Acceptance letter]

28 Aug 2024

PONE-D-24-18553R1 

PLOS ONE

Dear Dr. Cybulsky, 

I'm pleased to inform you that your manuscript has been deemed suitable for publication in PLOS ONE. Congratulations! Your manuscript is now being handed over to our production team.

Kind regards, 

on behalf of

Dr. Chen-Guang Liu 

Academic Editor

PLOS ONE